# Possible sarcopenia and depression among middle-aged and older adults in China: A 9-year longitudinal survey

**Xiang-yang He**[1][☉], **Zheng Liu**[1][☉], **Zhi-wei Lu**[2][☉], **Ren-cheng Zhao**[1][☉], **Yan-Fang Guo**[1][☉], **Qing Yuan**[1][☉], **Li Huang**[1][☉], **Xing-lin Zhong**[3][☉]*

**1** Department of Health Management, Shenzhen Baoan District Chronic Diseases Prevent and Cure Hospital, Shenzhen, Guangdong Province, China, **2** Department of Health Education, Guangdong Health Promotion and Education Center, Guangzhou, Guangdong Province, China, **3** Shenzhen Guangming District People's Hospital, Shenzhen, Guangdong Province, China

☉ Theese authors contributed equally to this work.
* 3314369279@qq.com

## Abstract

Possible sarcopenia (PS) and depression are prevalent among middle-aged and older adults. However, few studies have evaluated the causal association between depression and PS, as well as its components. This study conducted both cross-sectional and longitudinal analyses to explore the relationship between PS and depression in a population aged 45 and oledr. We evaluated the association between PS and its components with depression using data from the China Health and Retirement Longitudinal Study (CHARLS). PS was assessed according to the Asian Working Group for sarcopenia gudielines established in 2019 (AWGS 2019). Depression was measured by the validated 10-item Center for Epidemiological Studies Depression Scale (CES-D.10), with a cut-off score of 12 or higher indicating the presence of depression. 10,058 participants included in cross-sectional study and 5,726 participants without depression from the same cohort in 2011 were followed through 2020. Logistic regression and Cox proportional hazards models were employed to assess the association between PS and its components with depression. Restricted cubic spline (RCS) model was utilized to evaluate dose-response relationshipbetween muscle strength and physical performance with depression, and subgroup analyses were performed to validate the robustness of the findings. Cross-sectional analysis revealed that the prevalence of PS among middle-aged and older adults was 32.84% (3,303/10,058). Both PS (*OR*:1.47,95%*CI*:1.34–1.63), low muscle strength (LMS) (*OR*:1.46,95%*CI*:1.24–1.71) and low physical performance (LPP) (*OR*:1.45,95%*CI*:1.31–1.61) exhibited higher odds of depression after adjusting covariates. 1515 cases (26.46%) of incident depression were identified during the 9-years follow-up. Subjects with PS (*HR*:1.10,95%*CI*:1.01–1.19), LMS (*HR*:1.16,95%*CI*:1.01–1.34) and LPP (*HR*:1.08,95%*CI*:1.01–1.18) were at an elevated risk of new-onset depression compared to those without these conditions. The *RCS* analysis demonstrated a non-linear relationship between muscle strength and physical performance with depression (*p* > 0.05). Participants aged 50–59, married, with education below middle school, living in rural areas, non-smokers or non-drinkers, sleeping less than 8 hours, and classified as obese

**Data availability statement:** This study analyzed publicly available datasets. This data can be available here: https://charls.pku.edu.cn/. The data that support the findings of this study are available using the target URL: https://doi.org/10.5281/zenodo.14219812.

**Funding:** This study received support from the Medical Scientific Research Foundation of Guangdong Province of China (C2023107), Key Discipline of Chronic Non-communicable Disease Prevention and Control in Bao'an District, Shenzhen, Guangdong (2024JD232) and Key Medical Disciplines in Bao'an District (Prevention and Control of Chronic Non-communicable Diseases)

**Competing interests:** The authors have declared that no competing interests exist.

exhibited an increased risk in subgroup analysis. (all $p < 0.05$). PS, LMS and LPP were indentified as independent risk factors for new-onset depression. It is essential to assess muscle strength and physical performance in community-dwelling middle-aged and older adults using simple and feasible objective measures to enhance depression screening.

## Introduction

Sarcopenia is a skeletal muscle disorder characterized by a generalized and progressive decline in muscle strength and power. This condition can affect skeletal muscles throughout the body and increase the risk of adverse outcomes, such as fractures, falls, and mortality [1,2]. The causes and mechanisms of sarcopenia are not fully understood. However, several studies suggest that advanced age, low socioeconomic status, physical inactivity, poor dietary habits, being underweight, and the presence of comorbidities are significant risk factors for sarcopenia [1–3]. Early signs of sarcopenia have been observed in middle-aged and older adults, and these signs tend to escalate with advancing age [3]. China is one of the fastest-aging nations in the world. Investigating the prevalence of possible sarcopenia (PS) and sarcopenia among middle-aged and elderly adults to improve health and quality of life. Currently, the diagnosis of sarcopenia is not standardized globally, but usually includes an assessment of muscle mass, muscle strength or physical performance [4–6]. Diagnosing sarcopenia may require specialized equipment and techniques. The Asian Myasthenia Gravis Working Group (AWGS) introduced the concept of PS in 2019. This enables earlier detection and treatment of sarcopenia through simpler and more practical approaches in community screenings and clinical settings [5].

Depression is a complex chronic mental illness characterized by persistent low mood. It is strongly associated with disability, suicide and chronic illnesses (such as cardiovascular disease and diabetes) [7,8]. The rising prevalence of depression has become a significant public health problem due to escalating stress level. There are more than 54 million people with depression in China according to WHO report, and representing approximately 17% of the global burden of mental disorders [9]. A 2015 survey revealed that the prevalence of depression in the Chinese population aged ≥ 45 years was 37% [10], while in the elderly (aged > 60 years) ranged from 11% to 57% [11]. Depression is the most prevalent mental illness among middle-aged and elderly adults.

Sarcopenia and depression are prevalent disorders among middle-aged and elderly adults, often leading to adverse outcomes such as falls, disability, and chronic diseases [12,13]. Some studies have illustrated a complex relationship between sarcopenia and depression across various populations. This relationship may be attributed to shared common pathogenic factors, primarily including insufficient physical activity, nutritional deficiencies, chronic inflammation, and hormonal imbalances [14]. Sarcopenia can exacerbate depression through factors such as frequent falls, loss of independence, disruption of personal care, reduced nutritional intake, and decreased physical activity. Similarly, the prevalence of sarcopenia is higher in individuals with depression compared to the general population [12,13]. A cross-sectional study conducted in Brazil revealed that depression increased the risk of sarcopenia by 2.23 times, although it did not establish an association with PS [15]. Conversely, another study demonstrated a positive correlation between PS and depression [16]. However, a study focusing on the Asian population found no significant connection between sarcopenia and depression [17].

The majority of studies have been conducted on older populations with limited sample sizes and mostly cross-sectional studies. Therefore, it is crucial to examine the causal

relationship between sarcopenia and depression. This study aims to analyze the causal relationship between PS and its components with depression among middle-aged and older adults, using data from the China Health and Retirement Longitudinal Study (CHARLS) cohort conducted from 2011 to 2020.

## Methods

### Study population

This study is a secondary analysis of data from continuous longitudinal survey of CHARLS that a nationally representative cohort survey of subjects aged over 45 years in China. In 2011, CHARLS employed a multi-stage proportional stratified sampling process and a total of 17,708 respondents were randomly surveyed from 10,257 households in 450 representative villages or cities selected randomly from 150 counties or regions in 28 provinces in China. Subsequently, structured one-to-one interviews were conducted with study participants every 2 years to gather information on sociodemographics, lifestyle factors, and health-related data. Since its initiation, the project has conducted five follow-up visits in 2011 (baseline), 2013, 2015, 2018, and 2020. Additionally, CHARLS has received approval from the Medical Ethics Committee of Peking University and all participants provided informed consent prior to participation (IRB00001052–11015). Detailed methodology and core questionnaire of the project have been previously documented in other studies [18].

We utilized the baseline survey information from CHARLS 2011 in addition to the follow-up data from 2013 to 2020. The criteria for participant selection were (1) Age ≥ 45 years; (2) Information with complete determination of PS status. Exclusion criteria were (1) Missing important indicators such as age and sex; (2) Age < 45 years; (3) No CES-D.10 scores at baseline and follow-up assessments. Notably, we conducted distinct analyses of the cross-sectional and cohort data as distinct components in study.

### Data types

**Assessment of possible sarcopenia (PS).**  PS is defined as the presence of decreased muscle strength and/or reduced physical performance according to the AWGS 2019 consensus [5]. Muscle strength was evaluated based on the subject's maximum grip strength value. Mechanical grip force gauge WL-1000 was utilized to assess muscle strength in the study. Subjects were instructed to stand with their elbows at a right angle and squeezed the grip as hard as possible for a few seconds [5]. For individuals unable to stand, the test was conducted while seated. Both the left and right hands were tested twice and the final grip strength was determined by averaging the maximum grip strength of both hands. The diagnostic thresholds for muscle strength indicating PS are < 28 kg for men and < 18 kg for women [5].

Physical performance was assessed based on the subject's five sit-up times. Subjects were asked to sit on a 47 cm high stool with arms crossed over their chest, stand up as fast as possible and then complete next 5 repetitions and a total time taken was recorded. A cutoff value indicating a decline in physical performance recommended by AWGS 2019 is a 5-repetition sit-up time of ≥ 12 seconds [5,19].

**Assessment of depressive symptoms.**  The 10-item short form version of the Center for Epidemiological Studies Depression Scale (CES-D.10) was utilized in the CHARLS to assess depression [20]. This tool is well-established for screening depression in community populations and has demonstrated strong validity within the Chinese context. The CES-D.10 consists of 10 items, each rated on a scale of four variables, with possible scores ranging from 0 to 3 [21]. Notably, the 5th and 8th items are reverse-coded to enhance accuracy. Individuals who achieve a total score of 12 or higher across all 10 items are identified as exhibiting symptoms of depression in the study [22].

**Potential confounders.** The study adjusted for potential confounders, including sociodemographic characteristics and health-related factors. Sociodemographic characteristics included gender, age (45–49 years, 50–59 years, 60–69 years, and ≥70 years), residence (urban or rural), marital status (married or other, including divorced, widowed or unmarried), and education level (elementary school and below, secondary school, and tertiary education or above). Health-related factors considered incluede smoking, alcohol consumption, sleep duration (<6 hours, 6–8 hours, >8 hours), afternoon napping, and various chronic diseases (hypertension, hyperglycemia, dyslipidemia, heart disease, stroke, arthritis, cancer, and memory-related diseases). Body Mass Index (BMI) was calculated as body weight (*kg*) divided by the square of height (*m*). Weight was measured using Omron™ HN-286 scale, and height was measured using Seca™ 213 height meter. BMI was categorized into three groups according to the World Health Organization (WHO) BMI classification standard for Chinese individuals: BMI < 23.9 kg/m² for the underweight or normal weight group, BMI 24–27.9 kg/m2 for the overweight group, and BMI ≥ 28.0 kg/m2 for the obese group [23].

## Statistical analysis

The baseline characteristics and depressive symptoms from 2011 were described and compared in the study between participants with and without PS. Continuous variables that followed a normal distribution were presented as means ± standard deviation (SD). The independent samples *t*-test was employed to compare these variables between the two groups. Continuous variables that did not conform to normal distribution were reported as medians (percentiles), with comparisons between the two groups conducted using the *Kruskal-Wallis* test. For categorical variables, Which were represented as frequencies and percentages, the *chi-square* test was utilized for group comparisons. Additionally, *logistica* regression models were utilized to examine the association between PS and depression in the 2011 baseline data while controlling for sociodemographic characteristics and health-related factors. Furthermore, the relationship between muscle strength, physical performance, and depression were further explored.

To conduct longitudinal data analyses of follow-up, the duration of follow-up was measured based on the baseline and corresponding follow-up dates. Confounding factors, such as sociodemographic characteristics and health-related variables, were taken into account. Hazard ratios (*HR*) and 95% confidence intervals (*CI*) were calculated to assess the association between PS, low muscle strength (LMS), and low physical performance (LPP) with depression utilizing *COX* proportional hazards models. The study implemented four models to comprehensively evaluate the relationships. Model 1 was unadjusted, while Model 2 was adjusted for age, sex, education, marital status, and residence. Subsequently, Model 3 included smoking and alcohol consumption based on Model 2, along with sleep duration and afternoon napping. Building upon Model 3, Model 4 further incorporated BMI and several chronic diseases for a more robust analysis. Additionally, the study employed a restricted cubic spline (*RCS*) model to evaluate the dose-response relationship between muscle strength, physical performance, and the risk of new-onset depression. The study also examined the relationships between PS, muscle strength, and physical performance with depression across various demographic and health-related variables. All statistical analyses were conducted using IBM SPSS (Version 26.0) and R (Version 4.1.0, R Foundation for Statistical Computing). A *p-value* of less than 0.05 was considered statistically significant in this study.

## Result

### Study population

We enrolled a total of 10,058 participants for baseline characterization from the 17,708 respondents in the baseline survey conducted in 2011. Exclusions were based on the following

criteria: individuals with no information regarding age (n = 370), missing height or weight measurements (n = 1,558), missing data on the PS (n = 4,302), and missing data on at least one item in the CES-D.10 scale (n = 1,420). For the cohort analysis, we further excluded respondents who exhibited depression at baseline (n = 3,887) and those with missing CES-D.10 scores during the follow-up period (n = 445). A total of 5,726 subjects were included for cohort analysis. Fig 1 illustrates the detailed process for the inclusion of study subjects (Fig 1).

## Baseline characteristics of study population

The mean age of the study population was 59.33 ± 9.20 years, with 51.79% identifying as female and 8.63% residing in urban areas. Furthermore, 68.02% had an education level of elementary school or below, and 31.66% were current smokers. Among the 10,058 individuals surveyed, 32.84% were identified as having PS. The distribution of baseline characteristics for the surveyed population, both with and without PS, is presented in Table 1. Notably, the prevalence of PS was found to be higher among older individuals, those with a higher BMI, females, unmarried individuals, residents of rural areas, individuals with lower literacy levels, and those with comorbidities such as hypertension, diabetes, heart disease, stroke, memory-related diseases, rheumatism, and depression (all $P < 0.05$) (Table 1).

## Relationship between PS and depression at baseline

In the cross-sectional survey, S1 Table and Table 2 presents a positive correlation between PS, LMS, LPP, and depression. Among the subjects, 8.59% (864/10,058) exhibited LMS, while 29.12% (2,929/10,058) reported LPP. After adjusting for covariates such as age, sex, education, marital status, area of residence, smoking habits, alcohol consumption, sleep duration, afternoon napping, BMI, and chronic diseases, the analysis revealed that individuals with PS had a 47% higher risk of depression. Similarly, individuals with LMS and LPP demonstrated a 46% and 45% increased risk of depression, respectively, compared to those without these conditions (S1 Table and Table 2).

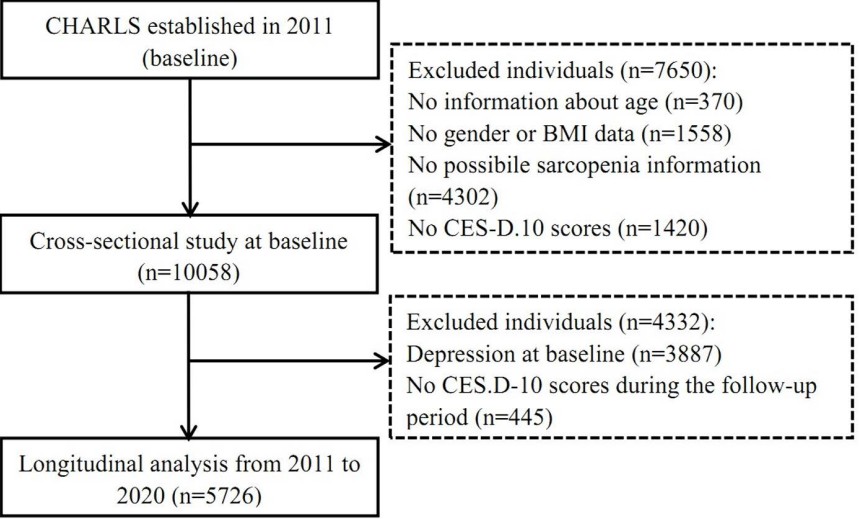

**Fig 1. Flowchart of participants in a cross-sectional and longitudinal study.**

**Table 1. Comparison of baseline characteristics between PS and normal population.**

| Characteristics | Overall (n = 10,058) | Normal control (n = 6,755) | PS group (n = 3,303) | *P-value* |
|---|---|---|---|---|
| Demographic | | | | |
| Age, years | 59.33 ± 9.20 | 57.51 ± 8.26 | 63.05 ± 9.90 | <0.001 |
| BMI, kg/m$^2$ | 23.44 ± 3.77 | 23.56 ± 3.65 | 23.17 ± 4.00 | <0.001 |
| Waistline,cm | 84.19 ± 12.53 | 84.25 ± 11.97 | 8,4.05 ± 13.60 | 0.462 |
| Female, *n*(%) | 5,209 (51.79) | 3,293 (48.75) | 1,916 (58.01) | <0.001 |
| Married, *n*(%) | 8,839 (87.88) | 6,116 (90.54) | 2,723 (82.44) | <0.001 |
| Socioeconomic status, *n*(%) | | | | |
| Urban residence, *n*(%) | 868 (8.63) | 641 (9.49) | 227 (6.87) | <0.001 |
| Education level, *n*(%) | | | | <0.001 |
| Primary school and below | 6,841 (68.02) | 4,234 (62.68) | 2607 (78.93) | |
| Middle school | 2,114 (21.02) | 1,617 (23.94) | 497 (15.05) | |
| High School or above | 1,103 (10.97) | 904 (13.38) | 199 (6.02) | |
| Health behavior, *n*(%) | | | | |
| Smoking, *n*(%) | | | | <0.001 |
| No | 5,972 (59.38) | 3,886 (57.53) | 2,086 (63.15) | |
| Yes | 3,184 (31.66) | 2,265 (33.53) | 919 (27.82) | |
| Quit | 902 (8.97) | 604 (8.94) | 298 (9.02) | |
| Alcohol consumption, *n*(%) | | | | <0.001 |
| Never | 6,684 (66.45) | 4,259 (63.05) | 2,425 (73.42) | |
| Less than once a month | 819 (8.14) | 605 (8.96) | 214 (6.48) | |
| More than once a month | 2,555 (25.40) | 1,891 (27.99) | 664 (20.10) | |
| Sleep duration,hours, *n*(%) | | | | <0.001 |
| ≤6 | 3,023 (30.06) | 1,896 (28.07) | 1,127 (34.12) | |
| 6–8 | 4,124 (41.00) | 2,909 (43.06) | 1,215 (36.78) | |
| >8 | 2,911 (28.94) | 1,950 (28.87) | 961 (29.10) | |
| With afternoon napping, *n*(%) | 4,614 (45.87) | 3,148 (46.60) | 1,466 (44.38) | 0.036 |
| Chronic diseases [a] | | | | |
| Hypertension, *n*(%) | 4,587 (45.61) | 2,870 (42.49) | 1,717 (51.98) | <0.001 |
| Dyslipidemia, *n*(%) | 7,324 (72.82) | 4,925 (72.91) | 2,399 (72.63) | 0.769 |
| Diabetes, *n*(%) | 1,334 (13.26) | 847 (12.54) | 487 (14.74) | 0.002 |
| Cancer, *n*(%) | 92 (0.92) | 62 (0.92) | 30 (0.91) | 0.975 |
| Heart disease, *n*(%) | 1,176 (11.75) | 659 (9.80) | 517 (15.76) | <0.001 |
| Stroke, *n*(%) | 197 (1.96) | 87 (1.29) | 110 (3.34) | <0.001 |
| Memory-related disease, *n*(%) | 120 (1.20) | 58 (0.86) | 62 (1.88) | 0.035 |
| Arthritis or rheumatism, *n*(%) | 3,458 (34.44) | 2,224 (32.98) | 1,234 (37.42) | 0.025 |
| Health self-assessment | | | | |
| CESD score, *n*(%) | 8.00 (4.00, 12.00) | 7.00 (4.00, 11.00) | 9.00 (5.00, 15.00) | <0.001 |
| Depression, *n*(%) | 3,887 (38.65) | 2,286 (33.84) | 1,601 (48.47) | <0.001 |
| Health self-assessment, *n*(%) | | | | <0.001 |
| Good | 4,750 (47.23) | 3,450 (51.07) | 1,300 (39.36) | |
| Fair | 3,721 (37.00) | 2,445 (36.20) | 1,276 (38.63) | |
| Poor | 1,587 (15.78) | 860 (12.73) | 727 (22.01) | |
| Handgrip strength (kg) | 32.00 (26.00, 32.00) | 34.50 (28.70, 42.00) | 27.00 (21.05, 34.35) | <0.001 |
| Time for 5 chair standing tests (*senconds*) | 10.72 ± 4.26 | 8.73 ± 1.88 | 14.80 ± 4.84 | <0.001 |

PS, possible sarcopenia.

[a]Missing data: 41 for cancer, 50 for heart disease, 20 for stroke, 25 for memory-related disease, 17 for arthritis or rheumatism.

**Table 2. Cross-sectional association of PS and its components with depression at baseline.**

| Variables | Case, n(%) | OR (95%CI) | | | |
|---|---|---|---|---|---|
| | | Model 1 | Model 2 | Model 3 | Model 4 |
| PS | 3,303(32.84) | 1.84(1.69,2.00) | 1.62(1.48,1.78) | 1.63(1.48,1.78) | 1.47(1.34,1.63) |
| LMS | 864(8.59) | 1.94(1.68,2.23) | 1.73(1.49,2.01) | 1.65(1.42,1.91) | 1.46(1.24,1.71) |
| LPP | 2,929(29.12) | 1.78(1.63,1.94) | 1.56(1.42,1.71) | 1.55(1.41,169) | 1.45(1.31,1.61) |

Model 1, no confounders were included.

Model 2, age gender, education, marital status and area of residence were included.

Model 3, smoking, alcohol consumption, sleep duration and afternoon napping into model 2.

Model 4, BMI and number of chronic diseases were added into model 3.

OR, odds ratio; CI, confidence interval; PS, possible sarcopenia; LMS, low muscle strength; LPP, low physical performance.

## Longitudinal association of PS and its components with depression

1,515 cases (26.46%) of new-onset depression were identified during the follow-up period from 2011 to 2020. S2 Table and Table 3 illustrates the association between subjects with PS at baseline and the incidence of new-onset depression. Subjects with PS exhibited a 21.0% higher risk of developing new-onset depression compared to those without the condition in the primary model. Furthermore, individuals with LMS and LPP had a 26.0% and 19.0% increased risk of new-onset depression, respectively. When covariates were considered, subjects with PS, LMS, and LPP demonstrated a 10%, 16%, and 8% increased risk of new-onset depression, respectively, in M odel 4 (S1 Table and Table 3).

## Subgroup analyses

Subgroup analyses were conducted across various demographics, including age, gender, marital status, area of residence, BMI, smoking status, alcohol consumption, and sleep duration. Fig 2 demonstrates that PS, LMS, and LPP significantly elevated the risk of new-onset depression in the 50–59 age group. However, these factors did not exhibit a significant association with the onset of depression in other age groups.

Fig 2 reveals that PS, LMS, and LPP may increase the risk of new-onset depression in rural populations. Analysis by gender indicated that LMS was associated with a higher risk of depression in males, while LPP showed no significant association with depression across genders. None of the PS components demonstrated a significant association with depression in females. Furthermore, LPP among obese individuals with a high BMI significantly elevated the risk of depression, whereas such an association was not observed in normal-weight or overweight populations (Fig 2).

**Table 3. Longitudinal association of PS and its components with depression.**

| Variables | Case, n(%) | HR (95%CI) | | | |
|---|---|---|---|---|---|
| | | Model1 | Model2 | Model3 | Model4 |
| PS | 1515(26.46) | 1.21(1.12,1.31) | 1.12(1.04,1.22) | 1.12(1.03,1.21) | 1.11(1.03,1.21) |
| LMS | 332(5.80) | 1.26(1.09,1.44) | 1.18(1.02,1.37) | 1.17(1.02,1.35) | 1.17(1.01,1.35) |
| LPP | 1346(23.51) | 1.19(1.09,1.28) | 1.09(1.00,1.18) | 1.08(1.00,1.18) | 1.09(1.01,1.19) |

Model 1, no confounders were included.

Model 2, age gender, education, marital status and area of residence were included.

Model 3, smoking, alcohol consumption, sleep duration and afternoon napping into model 2.

Model 4, BMI and number of chronic diseases were added into model 3.

HR, hazard ratio; CI, confidence interval; PS, possible sarcopenia; LMS, low muscle strength; LPP, Low physical performance.

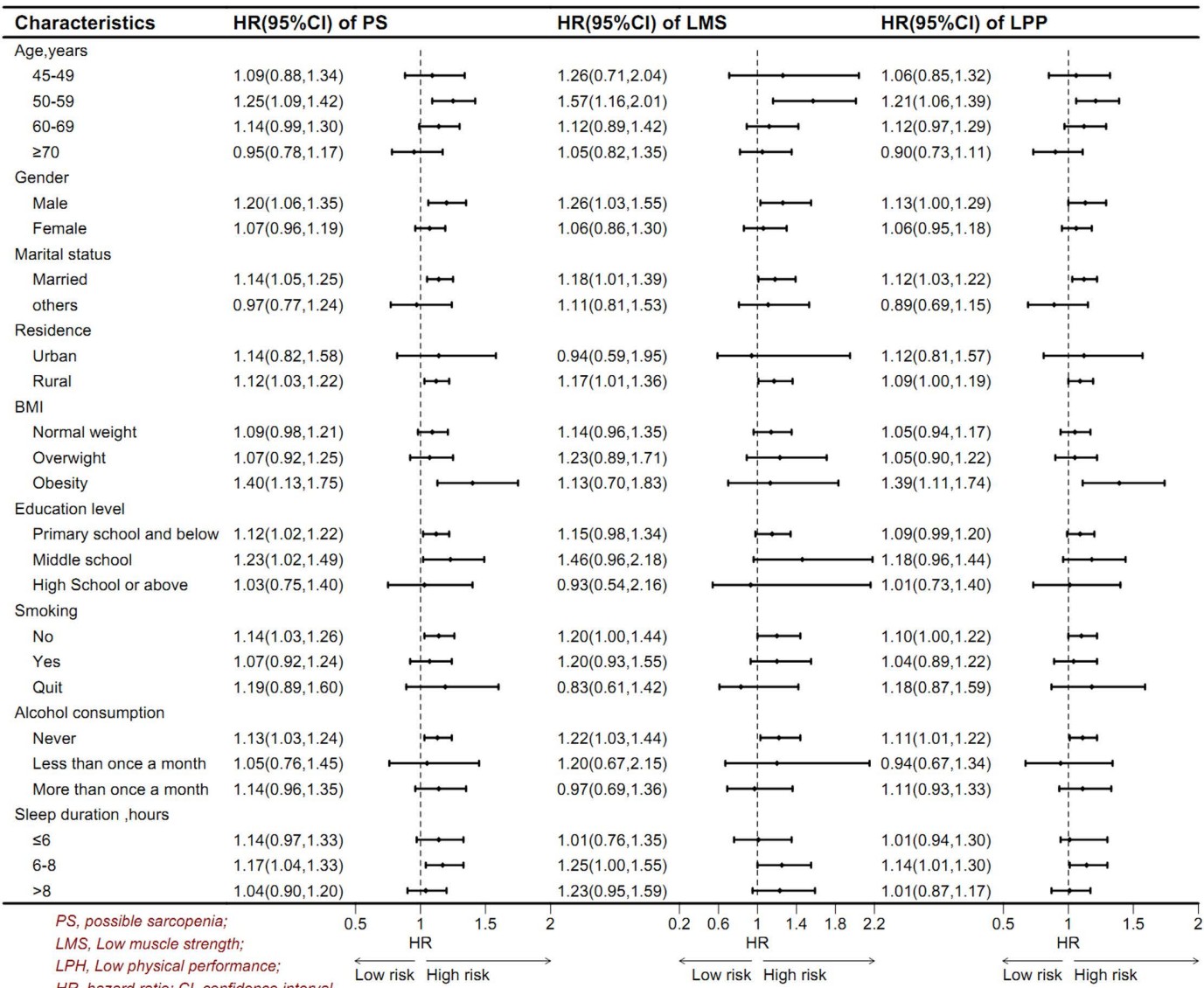

| Characteristics | HR(95%CI) of PS | | HR(95%CI) of LMS | | HR(95%CI) of LPP | |
|---|---|---|---|---|---|---|
| **Age,years** | | | | | | |
| 45-49 | 1.09(0.88,1.34) | | 1.26(0.71,2.04) | | 1.06(0.85,1.32) | |
| 50-59 | 1.25(1.09,1.42) | | 1.57(1.16,2.01) | | 1.21(1.06,1.39) | |
| 60-69 | 1.14(0.99,1.30) | | 1.12(0.89,1.42) | | 1.12(0.97,1.29) | |
| ≥70 | 0.95(0.78,1.17) | | 1.05(0.82,1.35) | | 0.90(0.73,1.11) | |
| **Gender** | | | | | | |
| Male | 1.20(1.06,1.35) | | 1.26(1.03,1.55) | | 1.13(1.00,1.29) | |
| Female | 1.07(0.96,1.19) | | 1.06(0.86,1.30) | | 1.06(0.95,1.18) | |
| **Marital status** | | | | | | |
| Married | 1.14(1.05,1.25) | | 1.18(1.01,1.39) | | 1.12(1.03,1.22) | |
| others | 0.97(0.77,1.24) | | 1.11(0.81,1.53) | | 0.89(0.69,1.15) | |
| **Residence** | | | | | | |
| Urban | 1.14(0.82,1.58) | | 0.94(0.59,1.95) | | 1.12(0.81,1.57) | |
| Rural | 1.12(1.03,1.22) | | 1.17(1.01,1.36) | | 1.09(1.00,1.19) | |
| **BMI** | | | | | | |
| Normal weight | 1.09(0.98,1.21) | | 1.14(0.96,1.35) | | 1.05(0.94,1.17) | |
| Overweight | 1.07(0.92,1.25) | | 1.23(0.89,1.71) | | 1.05(0.90,1.22) | |
| Obesity | 1.40(1.13,1.75) | | 1.13(0.70,1.83) | | 1.39(1.11,1.74) | |
| **Education level** | | | | | | |
| Primary school and below | 1.12(1.02,1.22) | | 1.15(0.98,1.34) | | 1.09(0.99,1.20) | |
| Middle school | 1.23(1.02,1.49) | | 1.46(0.96,2.18) | | 1.18(0.96,1.44) | |
| High School or above | 1.03(0.75,1.40) | | 0.93(0.54,2.16) | | 1.01(0.73,1.40) | |
| **Smoking** | | | | | | |
| No | 1.14(1.03,1.26) | | 1.20(1.00,1.44) | | 1.10(1.00,1.22) | |
| Yes | 1.07(0.92,1.24) | | 1.20(0.93,1.55) | | 1.04(0.89,1.22) | |
| Quit | 1.19(0.89,1.60) | | 0.83(0.61,1.42) | | 1.18(0.87,1.59) | |
| **Alcohol consumption** | | | | | | |
| Never | 1.13(1.03,1.24) | | 1.22(1.03,1.44) | | 1.11(1.01,1.22) | |
| Less than once a month | 1.05(0.76,1.45) | | 1.20(0.67,2.15) | | 0.94(0.67,1.34) | |
| More than once a month | 1.14(0.96,1.35) | | 0.97(0.69,1.36) | | 1.11(0.93,1.33) | |
| **Sleep duration ,hours** | | | | | | |
| ≤6 | 1.14(0.97,1.33) | | 1.01(0.76,1.35) | | 1.01(0.94,1.30) | |
| 6-8 | 1.17(1.04,1.33) | | 1.25(1.00,1.55) | | 1.14(1.01,1.30) | |
| >8 | 1.04(0.90,1.20) | | 1.23(0.95,1.59) | | 1.01(0.87,1.17) | |

*PS, possible sarcopenia;*
*LMS, Low muscle strength;*
*LPH, Low physical performance;*
*HR, hazard ratio; CI, confidence interval*

**Fig 2. Associations between PS, LMS and LPP with risk of new-onset depression in subgroup.**

## Dose-response relationship between possible sarcopenia and depressive symptoms

The relationships between muscle strength score and physical performance score and the incidence of depression were analyzed using RCS models based on the data from cohort study. Both muscle strength score and physical performance score exhibited a nonlinear dose-response connection with the risk of new-onset depression after correcting for confounders ($P_{-non-linear} < 0.05$). It suggests that as muscle strength increases, the risk of depression gradually decreases. Conversely, a decrease in physical performance status was associated with increase in the risk of depression (Fig 3, Fig 4).

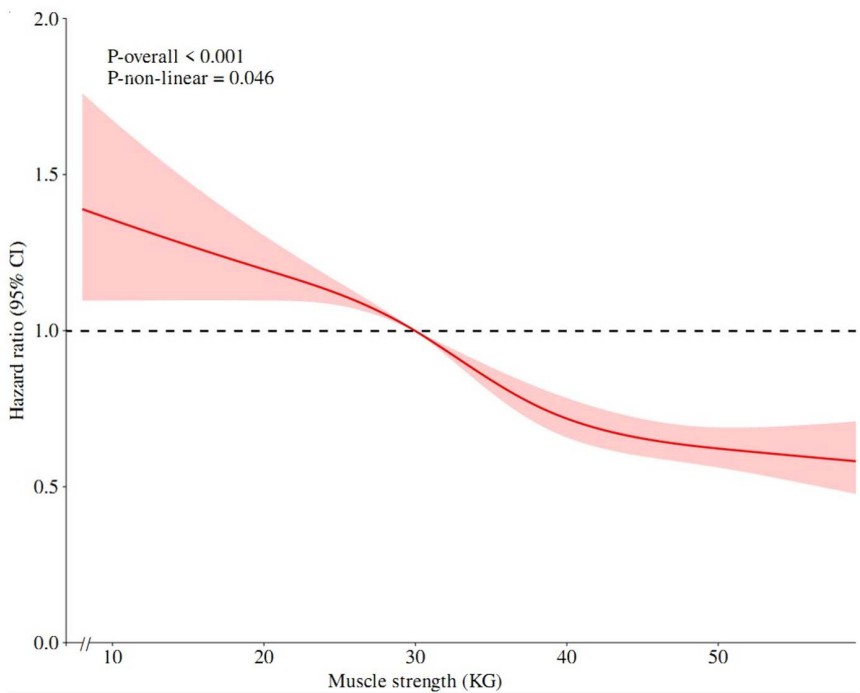

**Fig 3. Restricted cubic spline model of the association between muscel strength and depression.**

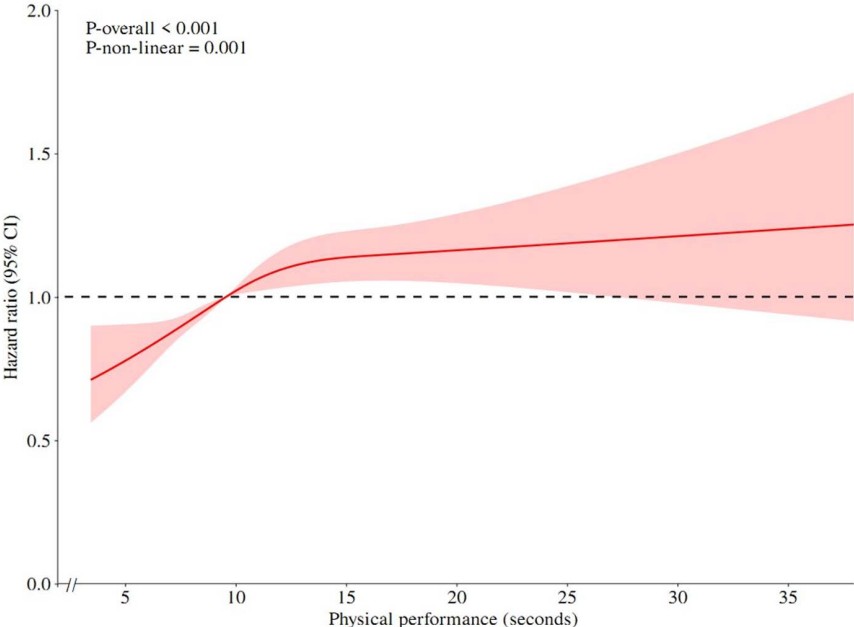

**Fig 4. Restricted cubic spline model of the association between physical performance and depression.**

## Discussion

The study utilizes data from the representative CHARLS. The association between PS and its components with the risk of depression among middle-aged and older adults is assessed for the first time based on data from both cross-sectional and cohort studies. The results revealed that PS, LMS and LPP significantly increased the risk of depression among middle-aged and older adults. Subgroup analyses indicated that the relationship between PS and its components with depression were not uniform across different population characteristics. It targeted interventions maybe necessary. Furthermore, results from RCS analysis hinted at a nonlinear relationship between muscle strength and physical performance with the risk of new-onset depression. Specifically, the risk of subsequent depression appeared to increase with greater muscle strength or declining physical performance. It emphasized the importance of maintaining these factors for mental well-being among the middle-aged and elderly population.

Most of the current studies on sarcopenia and PS have primarily focused on the elderly population, revealing varying prevalence rates reported due to differences in population characteristics [19,24]. One study indicated tha the prevalence of PS among middle-aged and older adults was 32.84%, which is higher than observed in Korea and Singapore [3,25]. Additionally, the prevalence of PS among the elderly was reported to be 38.5% and 46% in two other studies based on the CHARLS [26]. Despite some inconsistencies in the definition of sarcopenia across studies, the prevalence of PS is notably higher than that of sarcopenia [2,27,28]. This highlights the importance of early screening and diagnosis of PS in primary health care.

In cross-sectional surveys, individuals with PS exhibited significantly higher CES-D.10 scores than normal controls. Moreover, after adjusting confounders, the *OR* of individuals with PS, LMS and LPP were higher than those without those conditions. Longitudinal data analyses revealed a 1.10-times increase in the risk of new-onset depression among individuals with PS than normal controls. While the definition of PS varied across studies, the majority of findings suggest a strong association between PS and depression, which independent of population demographics [29-31]. Byeon et al. [17] found no significant association between PS and depression in a study among population aged over 20 years. The potential disparity in results may be attributed to variations in participant age and difference methods used to assess depression.

This study is the first to study causal relationship between PS and its components with depression in middle-aged and older Asian populations based on the representative data from cross-sectional and cohort studies. The results indicated that middle-aged and older adults with LMS and LPP were at a 1.16-times and 1.08-times increased risk of new-onset depression respectively. Several studies have pointed to the association between muscle strength and risk of depression according to the National Institutes of Health criteria [15,32]. A study had shown that LMS is linked to an elevated risk of depression [33]. Furthermore, LPP had been identified as exacerbating depression and increasing the risk of subsequent depression [34]. The mechanism underlying the connection between sarcopenia and depression is complex and involving common molecular pathways [35,36]. Nascimento CM et al. [37] and Hallgren M et al. [38] demonstrated that exercise can modulate hormone levels and reduce inflammation and decrease the risk of depression. Additionally, some researches has suggested that muscle atrophy can lead to chronic inflammatory responses, resulting in higher levels of inflammatory markers and an increased risk of depression [39,40]. The secretion of cytokines and peptides by skeletal muscle has been linked to improved brain function that can affect mood and cognition. Which suggests an interaction between muscle health and depression [41]. Estrogen disorders have been highlighted as a shared risk factor for sarcopenia and depression in other studies. It is need for further investigation into the mechanistic relationship between the two [42,43].

Various studies have demonstrated a complex relationship between muscle strength and depression across different populations. A study on community-dwelling older adults found a nonlinear correlation between muscle strength and depression [44]. Similarly, results from a survey of middle-aged and older Europeans suggested that higher muscle strength was linked to a decreased risk of depression [45]. Despite variations in regions and muscle strength categorizations, consistent outcomes reinforced the inverse relationship between muscle strength and depression. Physical performance also exhibited a nonlinear association with depression in our study. LPP was associated with a heightened risk of depression. It maybe due to LMS or LPP leading to physical inactivity and increased chronic inflammation [37–40]. Elevated inflammatory factors entering the brain could impact neurotransmitters, neuromodulators or neural circuits that can precipitate depression. Subgroup analyses can identify specific populations in need of targeted interventions. Therefore, it is critical to incorporate muscle strength and physical performance screening into depression prevention programs. Importantly, it is feasible to enhance muscle strength and physical performance through exercise training. Meta-analyses and cohort studies have underscored the benefits of aerobic and resistance training in enhancing muscle strength as well as ameliorating depression [46–48]. Thus the promotion of physical activity in middle-aged and elderly population in community can be effective in preventing sarcopenia and depression.

## Limitations

However, certain limitations of this study must be acknowledged. First, despite our adjustments for known confounders, it is important to recognize that additional confounders were not considered in our analysis. Second, CHARLS is an observational study, and the questionnaire survey methodology is susceptible to recall bias. Third, while the longitudinal study exhibited a stronger correlation between PS and depression, we were also unable to elucidate the underlying biological mechanisms driving this association. Consequently, further experimental research is essential to validate the relationship.

## Conclusion

The study concludes with new evidence suggesting a causal relationship between PS and its components with depression among Chinese individuals aged over 45 years. Enhancing physical activity and nutritional interventions may be beneficial for reducing and delaying the onset of depression.

## Supporting information

**S1 Table. Cross-sectional association of PS and its components with depression at baseline.**
(DOCX)

**S2 Table. Longitudinal association of PS and its components with depression.**
(DOCX)

**S1 Checklist. Plos-one-author-formatting-checklist.**
(DOCX)

## Acknowledgments

This study utilized data from the CHARLS. The authors extend their gratitude to the CHARLS research team, field staff, and participants for their contributions.

## Author contributions

**Conceptualization:** Zhi-wei Lu, Yan-Fang Guo, Qing Yuan, Xing-lin Zhong.

**Data curation:** Zheng Liu.

**Formal analysis:** Zhi-wei Lu, Yan-Fang Guo.

**Methodology:** Ren-cheng Zhao, Qing Yuan.

**Project administration:** Yan-Fang Guo.

**Software:** Ren-cheng Zhao, Li Huang.

**Supervision:** Qing Yuan, Xing-lin Zhong.

**Validation:** Li Huang.

**Visualization:** Li Huang.

**Writing – original draft:** Xiang-yang He, Zheng Liu.

**Writing – review & editing:** Xing-lin Zhong.

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
