## [Decision Letter · Decision Letter 0]

22 Nov 2024

PONE-D-24-42950Possible Sarcopenia and Depression among middle-aged and older adults in China: A 9-year Longitudinal SurveyPLOS ONE

Dear Dr. Zhong,

Thank you for submitting your manuscript to PLOS ONE. After careful consideration, we feel that it has merit but does not fully meet PLOS ONE’s publication criteria as it currently stands. Therefore, we invite you to submit a revised version of the manuscript that addresses the points raised during the review process.

**We have completed the review of your manuscript and a summary is appended below. The reviewer(s) have recommended some minor revisions to your manuscript.  Therefore, I invite you to respond to the reviewer(s)' comments and revise your manuscript.**

We look forward to receiving your revised manuscript.

Kind regards,

Sayani Das, PhD

Academic Editor

PLOS ONE

**Journal Requirements:**

3. Please note that your Data Availability Statement is currently missing the repository name. If your manuscript is accepted for publication, you will be asked to provide these details on a very short timeline. We therefore suggest that you provide this information now, though we will not hold up the peer review process if you are unable.

5. Please amend the manuscript submission data (via Edit Submission) to include author Dr. Zhi-wei Lu.

7. Please upload a new copy of Figure 2 as the detail is not clear. Please follow the link for more information: https://blogs.plos.org/plos/2019/06/looking-good-tips-for-creating-your-plos-figures-graphics/" https://blogs.plos.org/plos/2019/06/looking-good-tips-for-creating-your-plos-figures-graphics/

Reviewers' comments:

Reviewer's Responses to Questions

**Comments to the Author**

1. Is the manuscript technically sound, and do the data support the conclusions?

Reviewer #1: Yes

Reviewer #2: Yes

2. Has the statistical analysis been performed appropriately and rigorously? 

Reviewer #1: Yes

Reviewer #2: No

3. Have the authors made all data underlying the findings in their manuscript fully available?

Reviewer #1: Yes

Reviewer #2: Yes

4. Is the manuscript presented in an intelligible fashion and written in standard English?

Reviewer #1: Yes

Reviewer #2: Yes

5. Review Comments to the Author

**Reviewer #1:**  It was a pleasure to review the manuscript. The authors have used baseline and follow-up data from CHARLS cohort to investigate the association of possible sarcopenia (PS) and its components (low muscle strength or physical performance) with depression in both cross-sectional and longitudinal analysis. As expected, the study found a positive correlation between the two variates even after adjusting for potential confounders. Though the manuscript is well-written, there are few suggestions to improve the same.

• There are few grammatical errors (Examples – Page 3: Lines 39-41; Page 14: Lines 234-237) in the manuscript. The authors are advised to grammatically review the article and make corrections wherever applicable.

• Page 3: Line 44 – AWGS stands for Asian Working Group for Sarcopenia

• Page 4: Line 64 – Please use full expanded term for CHARLS when mentioned first time in the manuscript.

• Page 5: Line 89 – Please completely describe the handgrip dynamometer used for the device (spring-based or hydraulic). It is not clear whether WL-100 is the brand name or type of dynamometer. Please provide a complete description of the device so that the readers can easily find the dynamometer if they have to.

• Page 14: Line 227 - The authors have written that "..the risk of subsequent depression appeared to increase with “rising” muscle strength..". Please check and replace the word “rising” with another appropriate word (decreasing etc.).

**Reviewer #2:**  Manuscript ID: PONE-D-24-42950

Abstract: Structured Abstract. Written in a systematic way.

Introduction: Needs minor revision.

(1) Line 38-39 - Authors have only mentioned the effect of age as a potential factor for possible sarcopenia. Highlighting some existing literature that focused on other covariates like sex, socio-economic condition, marital status, health-related behavior etc. would give a better backdrop for the present study.

(2) Line 55-65 - The possible pathophysiological connection of simultaneous occurrence of PS and depression need to be briefly discussed at the introduction. Suggestive reading: Ji Eun Heo et al. (2018). Association between appendicular skeletal muscle mass and depressive symptoms: Review of the cardiovascular and metabolic diseases etiology research center cohort. Journal of Affective Disorders, 238, 8-15.

(3) No hypothesis or objective(s) are mentioned. Authors may create separate section for that.

Methods: Comprehensively written.

(1) Make a separate section 'Data Types' for Possible Sarcopenia, Depressive symptoms, and Confounders.

(2) Line 111 - What do you mean by 'Other' under the variable Marital Status. Please mention.

(3) Details of statistical analyses were systematically mentioned.

Results: Need major modification

(1) Table 2 and Table 3 should include the odds for the confounders as well, including the Model Summary (Adjusted R-square, Prediction Percentage etc.).

Discussion: No comments.

Overall comment: Recommending Acceptance after Minor Revision. Some grammatical errors are there in the manuscript. Authors should check that rigorously.

6. PLOS authors have the option to publish the peer review history of their article (what does this mean? ). If published, this will include your full peer review and any attached files.

**Do you want your identity to be public for this peer review?** For information about this choice, including consent withdrawal, please see our Privacy Policy .

Reviewer #1: **Yes: ** Dr. Sunny Singhal

Reviewer #2: **Yes: ** Dr. Akash Mallick

---

## [Author Response · Author response to Decision Letter 1]

27 Nov 2024

Dear Editor and Reviewers:

We appreciate the time and effort that you and the reviewers dedicated to providing feedback on our manuscript and are grateful for the insightful comments on and valuable improvements to our paper. We have incorporated most of the suggestions made by the reviewers. Please see below, in blue, for a point-by-point response to the reviewers’ comments and concerns. All page numbers refer to the revised manuscript file with tracked changes. Changes/additions to the manuscript are given in the red text. Those changes are highlighted in the manuscript.

Thanks very much for your attention to our paper.

Sincerely,

Xiang-yang He

Zheng Liu

Zhi-wei Lu

Ren-cheng Zhao

Yan-Fang Guo

Qing Yuan

Li Huang

Xing-lin Zhong

Journal Requirements:

We thank the reviewer for pointing this out. We have modified the paper (e.g., fonts, charts, references, etc.) according to the journal's requirements as much as possible. All changes are shown in red in the paper.

We were really sorry for our careless mistakes. Thank you for your reminder. We have modified the relevant information in the ‘Funding Information’ section as follows.

This study received support from the Medical Scientific Research Foundation of Guangdong Province of China (C2023107), Key Discipline of Chronic Non-communicable Disease Prevention and Control in Bao'an District, Shenzhen, Guangdong (2024JD232) and Key Medical Disciplines in Bao'an District (Prevention and Control of Chronic Non-communicable Diseases)

3. Please note that your Data Availability Statement is currently missing the repository name. If your manuscript is accepted for publication, you will be asked to provide these details on a very short timeline. We therefore suggest that you provide this information now, though we will not hold up the peer review process if you are unable.

Thank you for your reminder. We have uploaded the data to a database to ensure data accessibility as follows.

This study analyzed publicly available datasets. This data can be available here: https://charls.pku.edu.cn/. The data that support the findings of this study are available using the doi 10.5281/zenodo.14219811.

4.When completing the data availability statement of the submission form, you indicated that you will make your data available on acceptance. We strongly recommend all authors decide on a data sharing plan before acceptance, as the process can be lengthy and hold up publication timelines. Please note that, though access restrictions are acceptable now, your entire data will need to be made freely accessible if your manuscript is accepted for publication. This policy applies to all data except where public deposition would breach compliance with the protocol approved by your research ethics board. If you are unable to adhere to our open data policy, please kindly revise your statement to explain your reasoning and we will seek the editor's input on an exemption. Please be assured that, once you have provided your new statement, the assessment of your exemption will not hold up the peer review process.

Thank you for your reminder. We have revise the statement as follows.

This study analyzed publicly available datasets. This data can be available here: https://charls.pku.edu.cn/. The data that support the findings of this study are available using the doi 10.5281/zenodo.14219811.

5.Please amend the manuscript submission data (via Edit Submission) to include author Dr. Zhi-wei Lu.

Thank you for your reminder. We have amend the manuscript submission data (via Edit Submission) to include author Zhi-wei Lu.

6.Your ethics statement should only appear in the Methods section of your manuscript. If your ethics statement is written in any section besides the Methods, please delete it from any other section. 

Thank you for your reminder. We have removed the ethical statements except for the “Methods” section.

7. Please upload a new copy of Figure 2 as the detail is not clear. Please follow the link for more information:

https://blogs.plos.org/plos/2019/06/looking-good-tips-for-creating-your-plos-figures-graphics/" https://blogs.plos.org/plos/2019/06/looking-good-tips-for-creating-your-plos-figures-graphics/

We thank the reviewer for pointing this out. We have upload a new copy of figure 2.

 We thank the reviewer for pointing this out. We have updated reference list.

Dear Reviewers:

We appreciate the time and effort that you and the reviewers dedicated to providing feedback on our manuscript and are grateful for the insightful comments on and valuable improvements to our paper. We have incorporated most of the suggestions made by the reviewers. Please see below, in blue, for a point-by-point response to the reviewers’ comments and concerns. All page numbers refer to the revised manuscript file with tracked changes. Changes/additions to the manuscript are given in the red text. Those changes are highlighted in the manuscript.

Thanks very much for your attention to our paper.

Sincerely,

Xiang-yang He

Zheng Liu

Zhi-wei Lu

Ren-cheng Zhao

Yan-Fang Guo

Qing Yuan

Li Huang

Xing-lin Zhong

Review Comments to the Author

Reviewer #1: It was a pleasure to review the manuscript. The authors have used baseline and follow-up data from CHARLS cohort to investigate the association of possible sarcopenia (PS) and its components (low muscle strength or physical performance) with depression in both cross-sectional and longitudinal analysis. As expected, the study found a positive correlation between the two variates even after adjusting for potential confounders. Though the manuscript is well-written, there are few suggestions to improve the same.

• There are few grammatical errors (Examples – Page 3: Lines 39-41; Page 14: Lines 234-237) in the manuscript. The authors are advised to grammatically review the article and make corrections wherever applicable.

• Page 3: Line 44 – AWGS stands for Asian Working Group for Sarcopenia

• Page 4: Line 64 – Please use full expanded term for CHARLS when mentioned first time in the manuscript.

• Page 5: Line 89 – Please completely describe the handgrip dynamometer used for the device (spring-based or hydraulic). It is not clear whether WL-100 is the brand name or type of dynamometer. Please provide a complete description of the device so that the readers can easily find the dynamometer if they have to.

• Page 14: Line 227 - The authors have written that "..the risk of subsequent depression appeared to increase with “rising” muscle strength..". Please check and replace the word “rising” with another appropriate word (decreasing etc.).

We tried our best to improve the manuscript and made some changes to the manuscript. These changes will not influence the content and framework of the paper. And here we did not list the changes but marked in red in the revised manuscript with track changes. We appreciate for reviewers’ warm work earnestly and hope that the correction will meet with approval.

Reviewer #2: Manuscript ID: PONE-D-24-42950

Abstract: Structured Abstract. Written in a systematic way.

Introduction: Needs minor revision.

(1) Line 38-39 - Authors have only mentioned the effect of age as a potential factor for possible sarcopenia. Highlighting some existing literature that focused on other covariates like sex, socio-economic condition, marital status, health-related behavior etc. would give a better backdrop for the present study.

We sincerely appreciate the valuable comments. We conducted a thorough review of the literature and incorporated information regarding the impact of additional relevant factors, including variables such as advanced age, literacy, and exercise, on sarcopenia. This information has been modified on page 1, lines 65-70 in the revised manuscript as follows.

The causes and mechanisms of sarcopenia are not fully understood. However, several studies suggest that advanced age, low socioeconomic status, physical inactivity, poor dietary habits, being underweight, and the presence of comorbidities are significant risk factors for sarcopenia.

(2) Line 55-65 - The possible pathophysiological connection of simultaneous occurrence of PS and depression need to be briefly discussed at the introduction. Suggestive reading: Ji Eun Heo et al. (2018). Association between appendicular skeletal muscle mass and depressive symptoms: Review of the cardiovascular and metabolic diseases etiology research center cohort. Journal of Affective Disorders, 238, 8-15.

We sincerely appreciate the valuable comments. We conducted a thorough review of the literature and incorporated information. This information has been modified on page 3, lines 87-99 in the revised manuscript.

Sarcopenia and depression are prevalent disorders among middle-aged and elderly adults, often resulting in adverse outcomes such as falls, disability, and chronic diseases [12,13]. Various studies have demonstrated a complex relationship between sarcopenia and depression among different populations. This may be attributed to the presence of common pathogenic factors, primarily including insufficient physical activity, nutritional deficiencies, chronic inflammation, and hormonal imbalances[14]. Sarcopenia may contribute to depression due to factors such as frequent falls, loss of independence, interruption of personal care, reduced nutritional intake, and insufficient physical activity. Similarly, the prevalence of sarcopenia is higher in individual with depression compared to the general population [12,13]. A cross-sectional study in Brazil revealed that depression increased the risk of sarcopenia by 2.23 times, although it did not show an association with PS [15]. Conversely, another study demonstrated a positive correlation between PS and depression [16]. However, a study focusing on the Asian population found no significant connection between sarcopenia and depression [17].

(3) No hypothesis or objective(s) are mentioned. Authors may create separate section for that.

We sincerely appreciate the valuable comments. This information has been modified on page 3-4, lines 100-104 in the revised manuscript .

The majority of studies have been conducted on older populations with limited sample sizes and mostly cross-sectional studies. Therefore, it is crucial examine the causal relationship between sarcopenia and depression. This study aims to analyze the causal relationship between PS and its components with depression among middle-aged and older adults, using data from the China Health and Retirement Longitudinal Study (CHARLS) cohort conducted from 2011 to 2020.

Methods: Comprehensively written.

(1) Make a separate section 'Data Types' for Possible Sarcopenia, Depressive symptoms, and Confounders.

We appreciate for your warm work earnestly and hope that the correction will meet with approval. We have added a separate section “data type” , which contains data on Possible Sarcopenia, Depressive symptoms, and Confounders, see page 4-5, line 127-165.

Data Types

Assessment of Possible sarcopenia (PS)

PS is defined as the presence of decreased muscle strength and/or reduced physical performance according to the AWGS 2019 consensus [5]. Muscle strength was evaluated based on the subject's maximum grip strength value. Mechanical grip force gauge WL-1000 was utilized to assess muscle strength in the study. Subjects were instructed to stand with their elbows at a right angle and squeezed the grip as hard as possible for a few seconds [5]. For individuals unable to stand, the test was conducted while seated. Both the left and right hands were tested twice and the final grip strength was determined by averaging the maximum grip strength of both hands. The diagnostic thresholds for muscle strength indicating PS are <28 kg for men and <18 kg for women [5].

Physical performance was assessed based on the subject's five sit-up times. Subjects were asked to sit on a 47 cm high stool with arms crossed over their chest, stand up as fast as possible and then complete next 5 repetitions and a total time taken was recorded. A cutoff value indicating a decline in physical performance recommended by AWGS 2019 is a 5-repetition sit-up time of ≥12 seconds [5,19].

Assessment of Depressive symptoms

The 10-item short form version of the Center for Epidemiological Studies Depression Scale (CES-D.10) was utilized in the CHARLS to assess depression [20]. This tool is well-established for screening depression in community populations and has demonstrated strong validity within the Chinese context. The CES-D.10 consists of 10 items, each rated on a scale of four variables, with possible scores ranging from 0 to3 [21]. Notably, the 5th and 8th items are reverse-coded to enhance accuracy. Individuals who achieve a total score of 12 or higher across all 10 items are identified as exhibiting symptoms of depression in the study [22].

Potential confounders

The study adjusted for potential confounders, including sociodemographic characteristics and health-related factors. Sociodemographic characteristics included gender, age (45-49 years, 50-59 years, 60-69 years, and ≥70 years), residence (urban or rural), marital status (married or other) and education level (elementary school and below, secondary school, and tertiary education or above). Health-related factors considered incluede smoking, alcohol consumption, sleep duration (<6 hours, 6-8 hours, >8 hours), afternoon napping, and various chronic diseases (hypertension, hyperglycemia, dyslipidemia, heart disease, stroke, arthritis, cancer, and memory-related diseases). Body Mass Index (BMI) was calculated as body weight (kg) divided by the square of height (m). Weight was measured using OmronTM HN-286 scale, and height was measured using SecaTM 213 height meter. BMI was categorized into three groups according to the World Health Organization (WHO) BMI classification standard for Chinese individuals: BMI <23.9 kg/m2 for the underweight or normal weight group, BMI 24-27.9 kg/m2 for the overweight group, and BMI ≥28.0 kg/m2 for the obese group [23].

(2) Line 111 - What do you mean by 'Other' under the variable Marital Status. Please mention.

We sincerely appreciate the valuable comments. This information has been modified on page 5, lines 155-156 in the revised manuscript.

Sociodemographic characteristics included gender, age (45-49 years, 50-59 years, 60-69 years, and ≥70 years), residence (urban or rural), marital status (married or other,including divorced, widowed or unmarried)

(3) Details of statistical analyses were systematically mentioned.

Thank you again for your

---

## [Decision Letter · Decision Letter 1]

21 Jan 2025

Possible Sarcopenia and Depression among middle-aged and older adults in China: A 9-year Longitudinal Survey

PONE-D-24-42950R1

Dear Dr. Zhong,

We’re pleased to inform you that your manuscript has been judged scientifically suitable for publication and will be formally accepted for publication once it meets all outstanding technical requirements.

Kind regards,

Sayani Das, PhD

Academic Editor

PLOS ONE

Additional Editor Comments (optional):

Reviewers' comments:

Reviewer's Responses to Questions

**Comments to the Author**

1. If the authors have adequately addressed your comments raised in a previous round of review and you feel that this manuscript is now acceptable for publication, you may indicate that here to bypass the “Comments to the Author” section, enter your conflict of interest statement in the “Confidential to Editor” section, and submit your "Accept" recommendation.

Reviewer #1: (No Response)

Reviewer #2: All comments have been addressed

2. Is the manuscript technically sound, and do the data support the conclusions?

Reviewer #1: Partly

Reviewer #2: Yes

3. Has the statistical analysis been performed appropriately and rigorously? 

Reviewer #1: Yes

Reviewer #2: Yes

4. Have the authors made all data underlying the findings in their manuscript fully available?

Reviewer #1: Yes

Reviewer #2: Yes

5. Is the manuscript presented in an intelligible fashion and written in standard English?

Reviewer #1: Yes

Reviewer #2: Yes

6. Review Comments to the Author

Reviewer #1: It is very disheartening that the authors have chosen to disregard the earlier comments and have not made changes or provided reply to them. Please find the comments again for your perusal

1. Introduction - Paragraph 1: The authors have mentioned that the concept of PS (possible sarcopenia) was given by Asian Myasthenia Gravis Working Group (AWGS). However, the reference cited by author mention the group name as Asian Working Group for Sarcopenia. Kindly clarify.

2. Methodology - The authors have mentioned that they have used "Mechanical grip force gauge WL-1000" to assess muscle strength. However, AWGS 2019 guidelines clearly recommend the use of either the spring-type (Smedley) or hydraulic-type (Jamar) dynamometer. The reason for this is because the Asian cut-offs have been derived from the studies using these dynamometers only. As the grip strength measures varies as per the dynamometer used and the authors have not used either of the two recommended dynamometers, kindly provide justification how the AWGS 2019 cut-offs can be used to measure possible sarcopenia in this study.

3. Discussion - Paragraph 1: The authors have concluded that "the risk of subsequent depression appeared to increase with greater muscle strength...". However, in the earlier paragraph in the Results section subsection -"Dose-response relationship between Possible sarcopenia and depressive symptoms", the authors have reported that "It suggests that as muscle strength increases, the risk of depression gradually decreases.". These statements are contradictory to each other. Kindly clarify.

Regards

Reviewer #2: (No Response)

7. PLOS authors have the option to publish the peer review history of their article (what does this mean? ). If published, this will include your full peer review and any attached files.

**Do you want your identity to be public for this peer review?** For information about this choice, including consent withdrawal, please see our Privacy Policy .

Reviewer #1: **Yes: ** Dr. Sunny Singhal

Reviewer #2: **Yes: ** Dr. Akash Mallick

---

## [Editor Report · Acceptance letter]

PONE-D-24-42950R1

PLOS ONE

Dear Dr. Zhong,

I'm pleased to inform you that your manuscript has been deemed suitable for publication in PLOS ONE. Congratulations! Your manuscript is now being handed over to our production team.

Kind regards,

on behalf of

Dr Sayani Das

Academic Editor

PLOS ONE